# Neuroimmune Semaphorin 4A in Cancer Angiogenesis and Inflammation: A Promoter or a Suppressor?

**DOI:** 10.3390/ijms20010124

**Published:** 2018-12-30

**Authors:** Apoorva S. Iyer, Svetlana P. Chapoval

**Affiliations:** 1Center for Vascular and Inflammatory Diseases, University of Maryland School of Medicine, Baltimore, MD 21201, USA; apoorva.s.iyer@gmail.com; 2Department of Microbiology and Immunology, University of Maryland School of Medicine, Baltimore, MD 21201, USA; 3Program in Oncology at the Greenebaum Cancer Center, University of Maryland School of Medicine, Baltimore, MD 21201, USA; 4SemaPlex LLC, Ellicott City, MD 21042, USA

**Keywords:** neuroimmune semaphorins, plexins, cancer, angiogenesis, inflammation, VEGF, Sema4A

## Abstract

Neuroimmune semaphorin 4A (Sema4A), a member of semaphorin family of transmembrane and secreted proteins, is an important regulator of neuronal and immune functions. In the nervous system, Sema4A primarily regulates the functional activity of neurons serving as an axon guidance molecule. In the immune system, Sema4A regulates immune cell activation and function, instructing a fine tuning of the immune response. Recent studies have shown a dysregulation of Sema4A expression in several types of cancer such as hepatocellular carcinoma, colorectal, and breast cancers. Cancers have been associated with abnormal angiogenesis. The function of Sema4A in angiogenesis and cancer is not defined. Recent studies have demonstrated Sema4A expression and function in endothelial cells. However, the results of these studies are controversial as they report either pro- or anti-angiogenic Sema4A effects depending on the experimental settings. In this mini-review, we discuss these findings as well as our data on Sema4A regulation of inflammation and angiogenesis, which both are important pathologic processes underlining tumorigenesis and tumor metastasis. Understanding the role of Sema4A in those processes may guide the development of improved therapeutic treatments for cancer.

## 1. Introduction

Angiogenesis is a complex physiologic process which is tightly controlled by several proteins such as VEGF, FGF (fibroblast growth factor), PDGF (platelet-derived growth factor), angiopoietin-1 and -2, ephrin-B2, and others [1]. Under physiological conditions the blood vessels in adults are already formed and rarely branch or sprout [1,2]. However, when a blood vessel is damaged, a complex repair process is activated in which several types of cells and signals coordinate the functions of endothelial and muscle cells involved in repair [1,2,3]. There are important steps in angiogenesis: (1) protease production which includes matrix metalloproteases (MMPs), a desintegrin and metalloprotease domain (ADAMs), a desintegrin and metalloprotease domain with trombospondin motif (ADAMTs), cysteine proteases such as cathepsins, and serine proteases such as tissue plasminogen activator (tPA); (2) endothelial cell migration and proliferation; (3) vascular tube formation; (4) connections of newly formed tubes; (5) synthesis of a new basement membrane; and (6) incorporation of pericytes and smooth muscle cells [1,3]. In addition to pro-angiogenic stimuli named above, several angiogenesis inhibitors—such as angiostatin, endostatin, vasostatin, TIMP (tissue inhibitor of metalloproteinases), platelet factor-4, osteopontin, and others—halt angiogenesis by stopping a formation of new blood vessels or even promoting blood vessel removal [1,3]. These opposing stimuli tightly regulate vascular homeostasis.

Angiogenesis is also a vitally important process for tumor development and progression [3]. In order to grow, a tumor needs oxygen and nutrients which are supplied by new blood vessels. These vessels can form by an influence of angiogenic factors made by tumor cells themselves or by other surrounding cells which are stimulated by tumor cells to generate such factors [3]. The inhibitors of angiogenesis have been long considered as clinically important cancer-fighting agents. Most FDA (Food and Drug Administration) approved and clinically-useful anti-cancer therapeutics with anti-angiogenic effects are based on either inhibition or blockade of VEGF and its receptors. These include axitinib (tyrosine kinase inhibitor selective to VEGF-R1, -R2, -R3) for renal cell carcinoma [4]; bevacizumab (anti-VEGF humanized Ab) for several types of cancer, including lung, colorectal and cervical cancers [5]; sunitinib (triple-blocker, Abs to VEGF-R2, PDGF-Rb, and c-kit) for gastrointestinal stromal tumor, pancreatic, and renal cancers [6]; and several others [1,3]. However, more recent studies have shown some alarming side-effects in patients being treated with these drugs [7,8,9]. These undesirable consequences include an acute aortic dissection in a patient with liver tumors after a sixth round of sunitinib [7] or a jaw necrosis after axitinib treatment of a patient with renal cell carcinoma [9]. In some cases, the use of bevacizumab in patients with prostate cancer led to a confirmed anti-tumoral activity without a concomitant improvement in survival [8]. Moreover, targeting just one pathway in angiogenesis—e.g., VEGF—could be insufficient to disrupt cancer angiogenesis as other VEGF-unrelated pathways would stay intact. In addition to that, VEGF itself acting in tissues induces the expression of other molecules which can express either pro- or anti-angiogenic qualities thus promoting or compensating its direct effects. As an example of the above scenario, we previously have shown that the lung tissue VEGF expression induced a local inflammatory response characterized, in part, by a formation of new blood vessels, lung resident cell activation, and their upregulated expression of several neuroimmune proteins [10,11]. Among those upregulated proteins in lung DC (dendritic cells) was a member of Class IV semaphorin subfamily Sema4A. Thus, VEGF-induced lung tissue alterations can be, at least in part, Sema4A-mediated. However, whether Sema4A acts as an anti- or pro-angiogenic factor, remains to be determined as currently available publications examining its function came to opposite conclusions.

The previously published data have shown that Sema4A is preferentially expressed on DC and B cells in the immune system [12,13]. Sema4A has seven currently known receptors, Tim-2 (T cell, Ig domain, mucin domain-2) [14,15], NRP-1 (Neuropilin-1) [16], Plexin B1 [17], Plexin B2 [17], Plexin B3 [17], Plexin D1 [18], and most recently cloned ILT-4 [19]. The initial studies suggested that Plexin molecules are expressed on non-immune cells, whereas Tim-2 and ILT-4 expression was highly restricted to activated mouse and human CD4^+^ T cells, correspondingly [14,15,19], and NRP-1 expression was detected on mouse Treg cells [16]. However, later we and others demonstrated Plexin B1 and/or D1 expression in the immune system, particularly on DC [10,20], T cells [21] and Treg cells (our unpublished observations). As Sema4A regulates the immune response to different antigens such as allergens [19,22,23], infectious agents [14,21,24], and tissue-derived factors in autoimmunity [15,25,26,27,28], it is feasible to conclude that it plays a significant role in anti-tumor immunity that has not yet been assessed.

Sema4A-receptor pathways form a complex system of intracellular and extracellular signals which regulate different physiological and pathological tissue processes. For example, Sema4A regulates proper retina formation [29], correct guidance of hippocampal neurons [30], angiogenesis [18,31], and adaptive immune response [12,13,14,16,19]. On the other hand, the Sema4A pathways are dysregulated in different diseases such as retinal degenerative diseases (retinitis pigmentosa type 35 and cone-rod dystrophy type 10) [29], allergy [10,14,19,22,23], infectious [14,32] and autoimmune diseases [14,26,28], and certain types of cancer [16,33]. The individual impact of each Sema4A-receptor pair in disease pathogenesis and/or progression needs to be dissected separately for the whole picture of Sema4A impact to be envisioned. This could be done, first of all, in vitro by applying the receptor knock-out or specific receptor blocking techniques in cells of interest, and in vivo using individual Sema4A receptor-deficient mice and their inter-crosses in the experimental models of certain diseases.

## 2. Sema4A and Anti-Angiogenic Therapy in Cancer

Tumor progression and metastasis require a growth in local tumor angiogenesis where new blood vessels form in order to supply cancer cells with growth nutrients. Tumor cells themselves and tumor-associated stroma secrete angiogenesis-promoting factors such as angiopoietin-2, follistatin, G-CSF (granulocyte colony-stimulating factor), HGF (hepatocyte growth factor), IL-8 (Interleukin 8), leptin, PDGF-BB, PECAM-1, VEGF, and MMP-1, -2, -3, -7, -9, -10, -12, and -13 [34]. It has been shown that VEGF mRNA expression was mainly targeted to primary colorectal tumor cells whereas angiopoietin-2 and HGF mRNA expression was targeted to tumor-adjacent stromal cells [34]. Interestingly enough, recent studies have shown that many tissue-specific tumors can grow alongside the blood vessels without a formation of new ones [35], thus abating effects of anti-angiogenic therapies in such tumors. Nevertheless, several angiogenic factors such as VEGF-A, VEGF-B, angiopoietin-1, osteopontin, fibroblast growth factor, MMPs, and others currently serve as targets in cancer treatment with FDA-approved inhibitors which all are being used in conjunction with chemotherapy [1,3].

The main target for angiogenesis-based cancer therapy is VEGF [36]. Currently, there are several small molecule inhibitors and monoclonal antibodies targeting the VEGF-A pathway, with their side-effects analyzed and reported [36]. Bevacizumab (avastin, recombinant humanized monoclonal Ab to VEGF) is currently used for treatment of metastatic colorectal cancer, non-sqamous non-small cell lung cancer, glioblastoma, metastatic renal cell carcinoma, metastatic or recurrent cervical cancer (in combination with chemotherapy), platinum-resistant recurrent epithelial ovarian, fallopian tube, or primary peritoneal cancer in combination with chemotherapy. Cabozantinib (Cabometyx, Cometriq, a small molecule inhibitor of the tyrosine kinases c-Met and VEGFR2) and pazopanib (Votrient, an inhibitor of three VEGF receptors) are used to treat advanced renal carcinoma. This type of cancer is also treated by sorafenib (Nexavar, a small inhibitor of several tyrosine protein kinases, including VEGFR) which also demonstrates therapeutic effects toward un-resectable advanced hepatocellular carcinoma and progressive differentiated radioactive iodine-resistant thyroid carcinoma. More recently developed Zif-Aflibercept (Eylea, Zaltrap, VEGF-Trap, a hybrid fusion protein of VEGFR-1 and VEGFR-2 binding domains) is used for metastatic colorectal cancer that is resistant to an oxaliplatin-containing regimen.

The key side effect of anti-VEGF therapy is interference with the normal angiogenesis process where wound healing is highly disrupted (either delayed or incomplete). Indeed, when patients with metastatic colorectal carcinoma were treated with bevacizumab (Avastin, anti-VEGF-A mAb) they showed impaired wound healing and postoperative wound complications [37]. The reported side effects for bevacizumab include sensory neuropathy, hypertension, fatigue, and neutropenia [36]. Neutropenia and hypertension were also reported for ramucirumab use in addition to an increased risk of pneumonia. Thus, hypertension is the most known side-effect of VEGF inhibition as the ability of VEGF to decrease a blood pressure is well-documented. Other reported problems include an increased risk of arterial thromboembolic events caused by a disturbed regenerative capacity of endothelial cells [36].

As Sema4A is downstream of VEGF-induced signaling in the lung tissues [10,11], its effects on angiogenesis and tumor progression were of significant research interest. The role of Sema4A in angiogenesis has been previously evaluated in vitro and in vivo using either Sema4A-Fc fusion protein, recombinant human Sema4A, or/and Sema4A^−/−^ mice [18,38]. In developing mouse embryos, a co-expression of Sema4A and Plexin D1 in the intersomitic blood vessels was detected, suggesting the potential role of this ligand-receptor pair in vascular formation [18]. To evaluate such effect, the authors studied HUVEC (human umbilical vascular endothelial cells) migration in transwell chamber using VEGF alone or in combination with several semaphorins. They found that VEGF-induced cell migration was suppressed by Sema4A-Fc. Furthermore, Sema4A-Fc and Sema3E-Fc, but not Sema4D-Fc, inhibited VEGF-induced tubular structure formation by HUVEC in the in vitro angiogenesis assay (Figure 1). Interestingly enough, Sema4D showed the opposite to Sema4A effect on HUVEC, although these two semaphorins share Plexin D1 receptor [39,40]. This suggests the potential competition for the receptor binding between two semaphorins. However, it has been reported previously that the binding sites on Plexin D1 are different for each individual semaphorin ligand [40]. Nevertheless, there is a possibility that the binding of one Sema4 molecule could induce Plexin D1 modification, leading to another Sema4 molecule binding site to be hidden or inaccessible. Another study, however, has shown that a pro-angiogenic effect of Sema4D on endothelial cells is mediated by a different plexin family member, Plexin B1 [41], which is also a binding partner for Sema4A [38,39] (Figure 1). The signaling events occurring in endothelial cells under Sema4D exposure were dependent on a COOH-terminal PDZ-binding motif of Plexin B1, which binds two guanine nucleotide exchange factors for the small GTPase Rho, PDZ-RhoGEF and LARG, and were mediated by activation of Rho-initiated pathways. The signaling events under Sema4A exposure have never been examined in details.

The in vivo effect of rSema4A on vascularization in chick embryos has proven its indispensable role in blood vessel formation [18]. Chorioallantoic membrane (CAM) assays were used to evaluate such effects where gelatin sponges were inserted into chick embryos for three days. When examined thereafter, pre-treated with rSema4A sponges contained lower numbers of preformed blood vessels as compared to isotype control-pretreated sponges thus again proving the inhibitory role of Sema4A in angiogenesis. Pre-treatment of HUVEC with siRNA specific for individual Plexin family members, such as Plexin B1, D1, and A1, before rSema4A exposure determined Plexin D1 as its functional receptor on endothelial cells which mediates its anti-angiogenic activity (Figure 1) [18]. All of the discussed above results define Sema4A as a potent anti-angiogenic molecule and pave the way to its evaluation in cancer immunotherapy. However, a recent research by Meda and associates [31] has shown a pro-angiogenic role of Sema4A ligating Plexin D1 (Figure 1) on macrophages and stimulating their migration, VEGF-A production, and VEGF-R1 expression. Moreover, this Sema4A-VEGF-A pathway has been shown to be involved in macrophage activation and recruitment during inflammatory processes such as the experimental models of peritonitis and cardiac inflammation. Thus, considering the opposite effects of Sema4A on endothelial cells and macrophages, the identification of additional mechanisms of its action should be an important focus of future research aimed to develop of Sema4A-based therapeutic strategies to target cancer angiogenesis.

We previously reported that VEGF expression in lungs induces potent angiogenesis and edema formation [11]. Staining of mouse lung tissues with *Lycopersicon esculentum* lectin demonstrates a normal arrangement of blood vessels in the tracheas and intrapulmonary bronchi of wild-type mice. These blood vessels formed cascades with capillaries crossing between arterioles and venules. In contrast, we observed multiple endothelial sprouts, mostly arising from the venules, in VEGF transgenic mice as early as on day 3 of transgene expression induction. The vascular density (the percent of the airway covered with vessels) reached its maximum on day 7 and remained elevated for at least a month thereafter. The newly formed blood vessels were larger than the capillaries of the VEGF-unaffected control airways. The endothelial cells of these vessels were thin, had occasional fenestrations, and were enveloped by pericyte processes and basement membranes. Besides angiogenesis, we studied the effect of lung VEGF expression on local immune cells. We have shown that lung DC were activated by VEGF-A and upregulated of Sema4A and Plexin D1 expression [10]. Thus, for DC and macrophages, there is a positive feedback loop between VEGF-A and Sema4A which bind the corresponding receptors, Plexin D1 and VEGF-R1, and mediate this loop’s signaling pathways. However, as it has been shown previously and stated in the Introduction, Sema4A uses different receptors on different cell types to regulate their activation and function. For example, it uses Neuropilin-1 to mediate mouse Treg cell’s phenotype stabilization and function [16], Plexin B1 to induce such effect in human Treg cells (our unpublished observations), Tim-2 to co-stimulate mouse CD4^+^ T cells into Th1 phenotype [14,15], Plexin B2 for an optimal differentiation of CD8^+^ T cells [21], and ILT-4 to co-stimulate human CD4^+^ T cells into Th2 phenotype in vitro [19]. We did not detect Plexin B1 or Tim-2 expression on lung endothelial cells in mouse tissues either in steady-state or inflammatory conditions [10]. However, no such study was performed for human lung tissues. 

We analyzed the expression of Sema4A and Plexin D1 on human lung cancer tissue arrays using immunohistochemistry with corresponding Abs (Figure 2). We have found that blood vessels in cancer-associated inflammatory sights expressed both molecules (marked with red arrows on Figure 2). Thus, it is quite possible that Sema4A exerts its pro- or anti-angiogenic activity on pulmonary endothelial cells through Plexin D1 receptor. This statement, however, requires an extended focused testing.

Sema3A, which shares NRP-1 receptor with Sema4A, was identified as a potential anti-cancer semaphorin with anti-angiogenic signaling (Figure 1) [44]. Sema3A expression was analyzed in vitro in human cancerous cells and tissues and in vivo in three different genetically engineered mouse models of carcinogenesis [44]. The anti-tumor effects of Sema3A were directed toward the pruning and remodeling of abnormal blood vessels, and increasing their coverage with pericytes, all of which led to a stable vascular normalization. Based on these activities, Sema3A was termed ‘an endogenous angiogenesis inhibitor’. Moreover, the observed progressive decrease of Sema3A expression in endothelial cells starting from the pre-malignant lesions to actual tumors suggested its prognostic biomarker prospective for cancer progression.

Another potential therapeutic target in anti-angiogenic tumor management is Sema3G (Figure 1) which also shares NRP-1 receptor with Sema3A and Sema4A [45]. The transcriptomic profiling of different tissues linked Sema3G expression to endothelial cells during angiogenesis and development what led to its term “a vascular semaphoring”. Sema3G full length molecule (p87) has been shown to bind selectively NRP-2, whereas a processed by furin proprotein convertases molecule (p61) binds both, NRP-1 and -2. Unlike Sema3E, which inhibits angiogenesis by a NRP-independent binding of its receptor Plexin D1 [46], Sema3G-NRP-2 signaling positively affects angiogenesis [45] (Figure 1). NRP-1 is required as a co-receptor for VEGF165 signaling through its canonical tyrosine kinases VEGFR-1 and -2 [47,48]. Notably, NRPs serve as cell surface receptors for multiple semaphorin molecules with either pro- or anti-angiogenic effects. Therefore, currently existing and future cancer therapies based on the inhibition of individual components of the Sema-Plexin-NRP-VEGF complex have to take in account different cancer histotypes where distinct semaphorins and their either individual or cross-binding receptors are differently expressed and regulated.

Previously published data suggested that other Class III semaphorin members, Sema3B and Sema3F, could act as tumor suppressors as they bind antagonistically to NRP-1 and NRP-2, and inhibit angiogenesis (Figure 1) [49]. More detailed examination of their actions had supported a suppressive role for Sema3B in lung and renal cancers [50] and for Sema3F in oral squamous cell carcinoma [51].

We were interested in defining Sema4A’s effect on VEGF-induced lung vascularization and inflammation. The main question was whether Sema4A further deepens VEGF-induced lung pathologies acting as a pro-angiogenic and pro-inflammatory factor similarly to earlier described effects of fatty acid binding protein 4 (FABP4, adipocyte-FABP, aP2) [52], or whether it is produced as a compensatory protective molecule aimed to diminish or dampen VEGF-mediated tissue damages. FABP4 is an intracellular lipid chaperone which is induced in endothelial cells by VEGF exposure. It exhibits pro-angiogenic functions in vitro and in vivo by promoting endothelial cell proliferation, migration, survival, and morphogenesis. The generated VEGF tg/FABP4^−/−^ mice showed that FABP4 deficiency significantly reduced VEGF-induced airway angiogenesis and lung tissue inflammation.

## 3. Sema4A and Anti-Inflammatory Therapy in Cancer

We previously reported that lung VEGF-A expression induced local conventional DC (cDC) maturation and direction toward DC2 phenotype [11]. These VEGF-stimulated cDC upregulated Sema4A expression [10]. To assess the role of Sema4A in allergen-induced lung inflammation, we used OVA model of asthma in Sema4A^−/−^ mice where we found an exaggerated lung allergic response as compared to WT mice [22]. This suggests that Sema4A is a suppressive molecule for the in vivo Th2 response. We next crossed VEGF tg mice [11] with Sema4A^−/−^ mice [14] and have found that this semaphorin deficiency led to an increased inflammatory cell infiltration in the lungs of VEGF tg mice when transgene expression is turned on by doxycycline-containing water (Figure 3). As we have shown previously, lung bronchial epithelial expression of VEGF transgene lead to an asthma-like phenotype with inflammation, parenchymal remodeling, increased vascularization, edema formation, mucous cell and myocyte hyperplasia and airway hyperreactivity [11]. The observed lung tissue inflammatory response and vascularization in our VEGF tg/Sema4A^−/−^ mice were more pronounced than those found in transgenic mice alone (Figure 3). In addition, we observed higher local levels of Th2 cytokine IL-13 in VEGF tg mice with Sema4A deficiency (Figure 3B). In fact, IL-13 was a signature Th2 cytokine, in contrast to unchanged levels of IL-4 and IL-5, upregulated in the Sema4A^−/−^ lungs and spleens after allergen exposure [22]. Based on the well-established role of VEGF in angiogenesis and tumor pathogenesis, our preliminary data for the mouse models of experimental asthma in Sema4A^−/−^ and VEGF tg/Sema4A^−/−^ mice, and the discussed above publications on the Sema4A inhibitory role in VEGF-induced angiogenesis, we suggest that Sema4A may act as a tumor suppressor interfering at least with three critical pathways in tumor development, progression, and metastasis: (1) immune cell activation and function; (2) inflammation; and (3) angiogenesis.

Based on all of the above, Sema4A is a suppressive molecule for both, allergen-induced and VEGF-mediated lung tissue responses, making it an attractive target for allergic disease immunotherapy. Indeed, when recombinant Sema4A protein was introduced into the allergic murine lungs, it significantly suppressed all features of an inflammatory Th2 response such as lung eosinophilia, mucus hypersecretion, proinflammatory, and Th2 cytokine production [22]. We and others have shown that Sema4A affects Treg cells in vitro and in vivo [16,22] as Treg cell local lung number decreases under inflammation by Sema4A deficiency [22]. Moreover, Sema4A acting through NRP-1 in mice [16] and Plexin B1 in humans (our unpublished observation) stabilizes Treg cell number and function. Therefore, Sema4A serves as a downregulatory molecule for allergic diseases suppressing allergen-dependent and -independent responses, in part, by upregulating Treg cell response. However, a recent article by Lu and colleagues [19] has demonstrated a costimulatory effect of Sema4A for T cell, especially Th2 cell, activation and function. Further studies are warranted to elucidate Sema4A-ILT-4 roles in different diseases including cancer.

As a translational part of our research, we obtained human lung cancer tissue arrays (Z7020065, BioChain) and assessed them for Sema4A and corresponding receptor expression using commercially available antibodies. Tissue photomicrographs were taken using CoolSnap image capturing software (Roper Scientific Inc.) with Nikon Eclipse E400 (Japan) microscope. The tissue arrays constituted of: (1) adenocarcinoma, Stages I to III (Stage I: the tumor is only present in the lungs, Stage II: the cancer has invaded the lymph nodes, Stage III: the cancer has invaded other organs); (2) bronchioalveolar carcinoma; (3) papillary carcinoma; (4) squamous cell carcinoma; and (5) small cell lung cancer. We have found a low to absent Sema4A expression in bronchioalveolar, papillary, and small cell lung cancer but stage-dependent increased levels of Sema4A in adenocarcinoma and squamous cell carcinoma (Figure 4). This observation supports a previous notion that the effects of different semaphorins on cancer progression are broad and context-dependent [40,53,54].

Cumulative recent findings define Sema3A as a potent anti-angiogenic and anti-malignant molecule in different types of cancer [42,44,55,56]. Initial study defined Sema3A as an endogenous angiogenesis inhibitor which expression in cancerous tissues was gradually declined with disease progression [44]. Moreover, recently published research showed that a downregulation of Sema3A expression promoted cancer metastasis [42]. The latter study focused on non-small cell lung carcinoma (NSCLC) where miRNA-362-5p overexpression was associated with Sema3A downregulation as opposed to their expression levels in normal tissues. Demonstrated direct Sema3A–miRNA-362 interaction affected NSCLC invasion, migration and colony formation. This study suggests that Sema3A is a direct target gene for miRNA-362 (Figure 1) and it functions as angiogenesis and metastasis inhibitor. Another recent study associated multiple myeloma (MM) and leukemia with a low expression of Sema3A as compared to normal control [57]. Serum Sema3A concentration inversely correlated with the MM stage what makes this semaphorin molecule a prospective prognostic marker for a disease course. In parallel of a gradual replacement of healthy Sema3A-producing bone marrow cells with malignant cells, there was a concurrent increase in their VEGF expression what further complicated MM. The value of the Sema3A expression level in different types of cancer as a marker for a disease prognosis was discussed in a study designed to generate a safe tumor-suppressive Sema3A point mutant isoform [55]. For instance, this study showed that Sema3A binding NRP-1 can increase vascular permeability without an inhibition of tumor growth (Figure 1) [55]. Moreover, it demonstrated that main anti-angiogenic and anti-tumorigenic Sema3A activities were independent of NRP-1 binding but were the results of Sema3A-Plexin A4 signaling. Therefore, the efforts were made to design a Sema3A mutant which binds Plexin A4 but not NRP-1 [55]. Such a mutant, Sema3A_ Ig-b, was effective in vasculature normalization, tumor inhibition, slowing metastasis, and improving chemotherapy.

In contrast to Sema3A tumor suppressor effects, Sema3E, which shares Plexin D1 receptor with Sema4A and Sema4D, consistently demonstrated the pro-tumoral effects [40,46,58]. Sema3E-Plexin D1 interaction promoted tumor growth and metastasis (Figure 1) [46,58], and this ligand-receptor pair expression correlated positively with metastatic progression of colon, liver, and melanoma cancers [46]. Moreover, knocking down either Sema3E or Plexin D1 hampered a metastatic potential of several human cancer cells upon xenotransplantation indicating the importance of these molecules in metastatic process. Extensive analysis of Sema3E gene expression in human colon carcinomas demonstrated its 88% association with metastatic disease. Analysis of human breast cancer showed elevated Sema3E expression in metastatic breast cancer as well [58]. Sema3E knock-down in breast cancer cell lines triggered their apoptotic death which was rescued by rSema3E additions to cell cultures. The same effect was observed in cultures treated with a synthetic blocker of Sema3E-Plexin D1 interaction, SD1, which is a soluble recombinant protein containing Sema3E-binding Plexin D1 domain. Unexpectedly, two contrasting roles of Sema3E toward cancer have been demonstrated. It acts as an angiogenesis inhibitor by limiting tumor vessel density and as a metastasis promoter by stimulating tumor cell invasiveness, transmigration, and extravasation [46]. Interestingly, Sema3E molecule display Sema4A-like function in allergic asthma as an inflammation inhibitor [59,60]. Furthermore, similarly to the reported effects of recombinant Sema4A administrations to allergic lungs [61], intranasal instillations of Sema3E protected mice from allergen-induced airway inflammatory responses [59,60]. Therefore, certain functional parallels can be drawn between these two distinct semaphorins, Sema3E and Sema4A, but to this day it is unclear if their seeming functional likeness in some diseases—asthma, for example—is directly related to their signaling through Plexin D1.

Given the many potential impacts of Sema4A on tumors, its detailed investigation will be beneficial for basic and clinical cancer research.

## 4. Conclusions

The presented and discussed here data show that Sema4A functions as a VEGF-opposing molecule in lung inflammation, however, its role in tumorigenesis and metastasis is not clearly defined. The currently proven association of Sema4A mutation with Familial colorectal cancer type X (FCCTX) was first reported in 2014 [33,34]. A more recent study connected the increased Sema4A expression in breast cancer tissues with disease progression [62]. Surprisingly, no new data for other types of cancer has been shown since then. Considering multiple receptors translating Sema4A effects into different cells, the individual and/or dominating receptor function needs to be detected and analyzed first. Therefore, additional functions of Sema4A are likely to emerge in the near future.

## Figures and Tables

**Figure 1 ijms-20-00124-f001:**
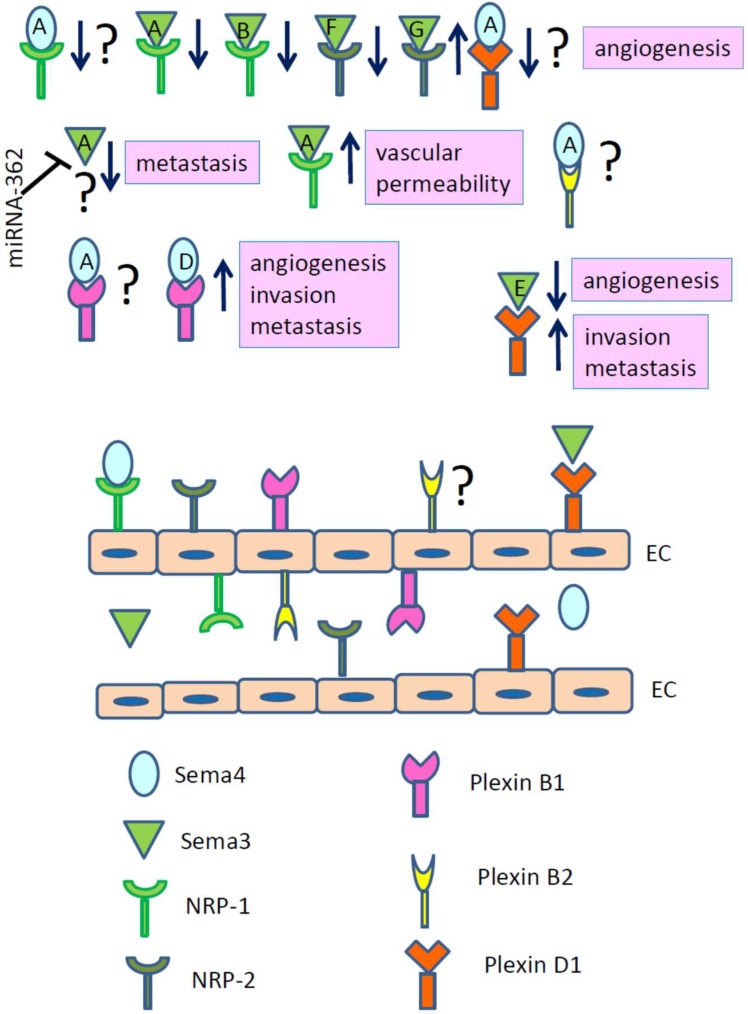
Semaphorins signaling by linking the corresponding receptors contribute to several hallmarks of cancer including, but not limited to, angiogenesis and metastasis. Sema4A binding NRP-1 receptor sends anti-angiogenic signal to endothelial cells (EC). The figure shows that Class III members of semaphorin family of proteins such as Sema3A, Sema3B, and Sema3F, inhibit angiogenesis upon interaction with corresponding NRP-1 or -2, whereas Sema3G-NRP-2 interaction leads to angiogenesis inhibition. Sema4A-Plexin D1 interaction demonstrated anti-angiogenic effects in vivo in one study [18]. Sema3A is a direct target gene for miRNA-362 and it functions as angiogenesis and metastasis inhibitor [42]. Sema3E-Plexin D1signaling also demonstrated anti-angiogenic effects, however, promoted cancer invasion and metastasis. Both class IV semaphorin molecules, Sema4A and Sema4D, functionally interact with Plexin B1, which is expressed on EC. Sema4D–Plexin B1 interaction has been shown to promote cancer-related angiogenesis, cancer invasion, and metastasis [41]. The role of Sema4A-Plexin B1 interaction in these processes is unclear. In addition to that, it is not known if Plexin B2, with which Plexin B1 forms a functional heterodimer [43], is also expressed and functions on EC. Currently existing and future cancer therapies based on the inhibition of individual components of the Sema-Plexin-NRP-VEGF complex have to take in account different cancer histotypes where distinct semaphorins and their either individual or cross-binding receptors are differently expressed and regulated.

**Figure 2 ijms-20-00124-f002:**
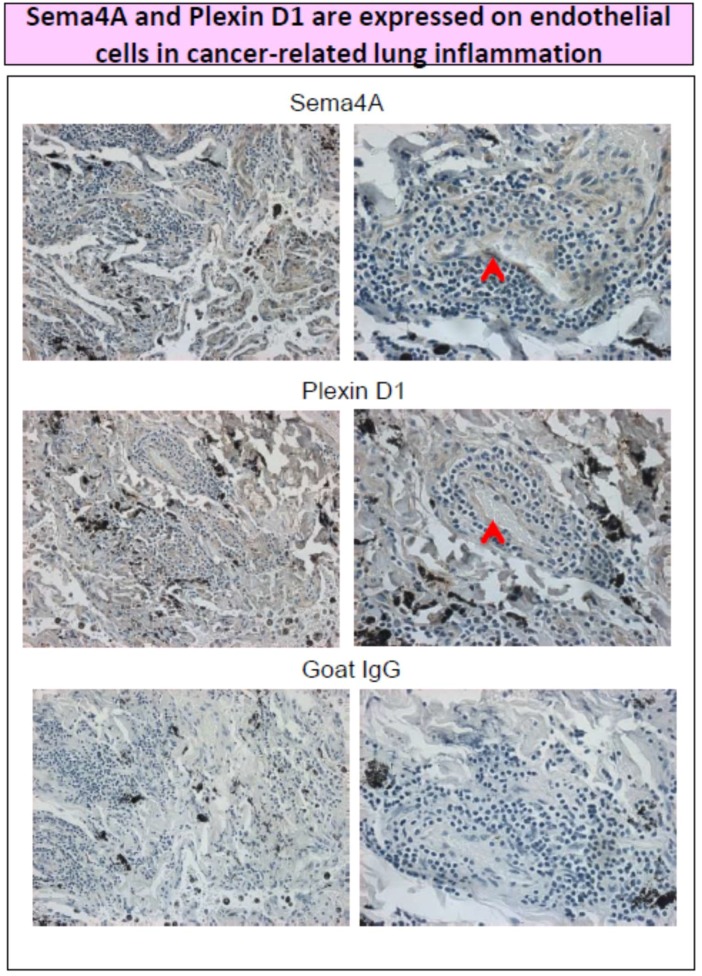
Immunohistochemistry of human lung cancer-adjoined tissue on BioChain arrays was performed as a four-step assay. Primary Ab for Sema4A (sc-46258) and Plexin D1 (E-13) were obtained from Santa Cruz Biotech. Streptavidin-HRP (Abcam) was used as a detection enzyme and DAB peroxidase substrate kit (SK-4100, Vector) was used for staining visualization. Biotynilated rabbit anti-goat IgG was used as a secondary Ab. Red arrows indicate marker expression on endothelial cells. Panels on the left represent ×20 magnification, panels on the right show a magnification of ×40.

**Figure 3 ijms-20-00124-f003:**
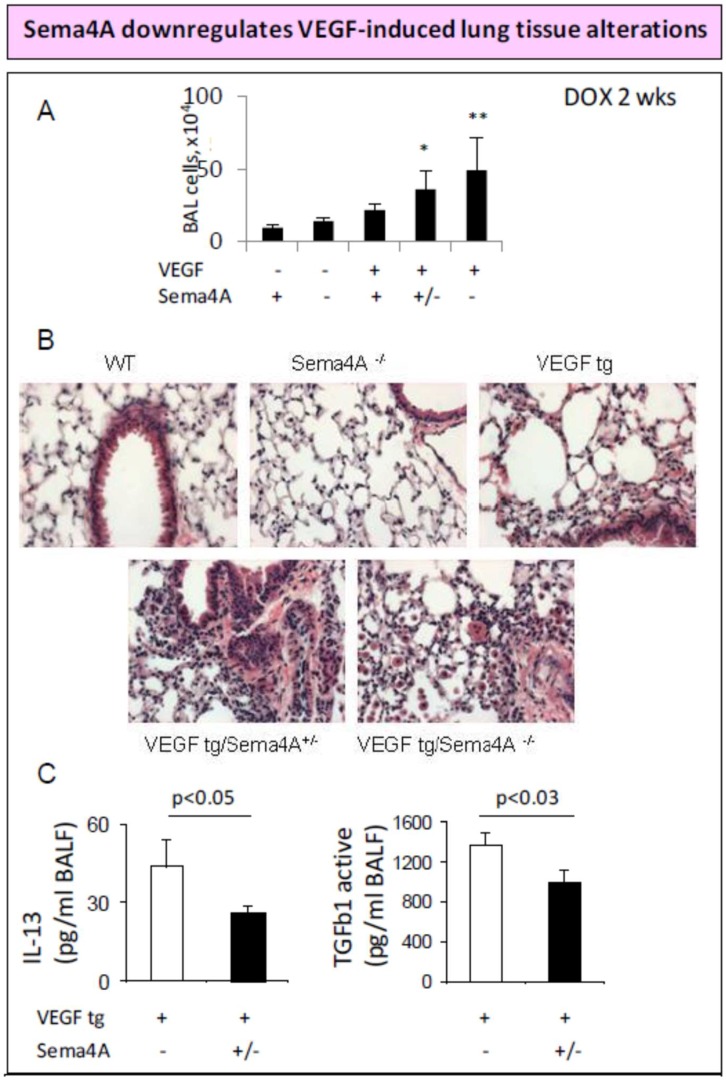
Increase in lung pathology in VEGF tg mice with Sema4A deficiency. (**A**) Broncho-alveolar lavage (BAL) fluids were harvested and the total cell counts were performed with hemacytometer. * *p* < 0.03, VEGF tg/ Sema4A^+/−^ mice vs. VEGF tg mice; ** *p* < 0.05, VEGF tg/Sema4A^−/−^ mice vs. VEGF tg mice. *N* = 3–4 animals/experimental group. (**B**) Lung sections were stained with H&E. Photomicrographs of 40× magnification H&E stained lung sections were taken. (**C**) IL-13 and TGFb1 levels in BAL fluids were measured by ELISA (Quantikine). Data represented as concentration of cytokine ± SEM.

**Figure 4 ijms-20-00124-f004:**
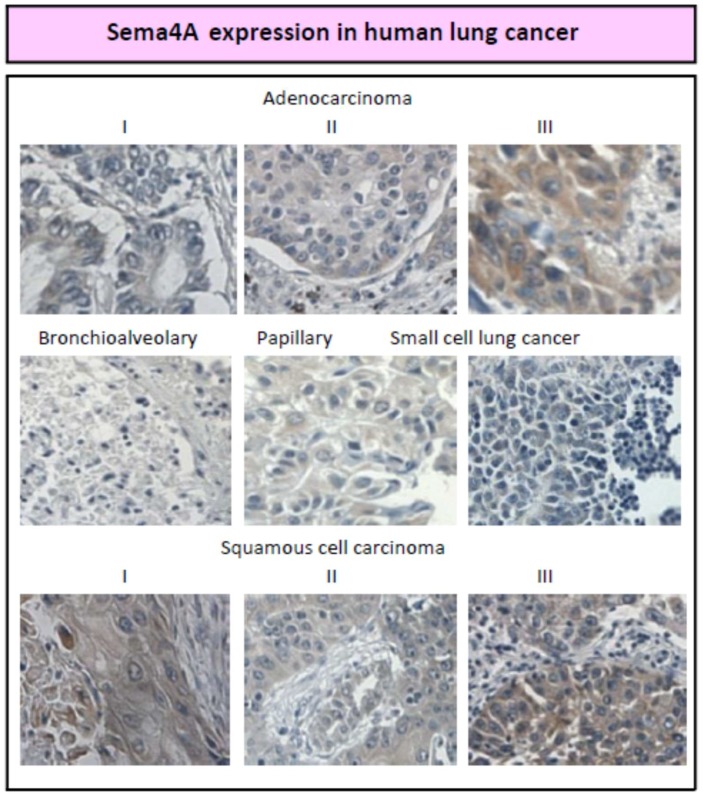
Immunohistochemistry of human lung cancerous tissue arrays (BioChain) was visualized via a four-step staining procedure using anti-Sema4A (sc-46258, Santa Cruz) as primary Ab, biotinylated rabbit anti-goat IgG (sc-2774) as secondary Ab, streptavidin-HRP (Abcam) as detection enzyme, and DAB peroxidase substrate kit (SK-4100, Vector) for visualization (100× magnification). The stages of cancer are shown in Roman numerals. Staining specificity was compared with goat IgG control Ab stain (not shown).

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
