# Peer review of "Neuroimmune Semaphorin 4A in Cancer Angiogenesis and Inflammation: A Promoter or a Suppressor?"

_ijms, 2018, doi:10.3390/ijms20010124_

Round 1

Reviewer 1 Report

Apoorva Iyer & Svetlana P. Chapoval describe very well in this review the role of Sema4A in the regulation of immune system and pathological angiogenesis, presenting both their own data and discussing findings deriving from the literature. This mixed review/article is well written and the authors made a comprehensive overview of the role of Sema4A in several pathological processes. The data the authors presented, the complex relationship between Sema4A and the different Plexins/Nrps receptors and in particular on the role of Sema4A in atsma, inflammation and allergic diseases, are interesting.

However there are some issues and concerns regarding the author’s statements and descriptions of data present in the literature, as well as regarding some of the interpretations of their results, that should be better addressed and modified before publication.

Main concerns:

1. In general along the text and in particular in the conclusions, the role (and “cross-role”) of other Semas in tumor angiogenesis and cancer is not properly described (or clearly), cited and interpreted. They have, at least, to be briefly and better describe it to better put Sema4A in the context of the “complex” system of cancer.

For instance, it has been widely described that different Sema3s (not all well cited in the text) are regulated in tumor angiogenesis, immunity and tumor progression: i) such as the pro-tumoral Sema3E (Luchino J et al Cancer Cell 2013; Casazza et al J Clin Invest 2010; Neufeld G et al, Cell Adh Migr. 2016); ii) Sema3G, as pro-angiogenic (Wallerius M et al, Arterioscler Thromb Vasc Biol 2011); iii) Sema3A, described as endogenous anti-angiogenic and anti-tumor molecule in different mouse model of cancers and human tumors (Luo D et al J Immunol Res. 2018, Gioelli et al Scie. Transl Med 2018, Maione et al., J Clin Invest 2009) and able to revert tumor-associated macrophages (Wallerius M, et al 2016, Cancer Res).

In particular, in the conclusion, when the authors describe some Sema3 as comparison to Sema4A, they should tak in account a more global scenario, to better described the opposite effects observed of Sema4A (and other Semas) in relationship to VEGF and other angiogenic factors.

The authors, in this part of the conclusion should add more appropriate references.

In addition, based on the recent literature, is not fully correct the statement of the rows “259-260” in which the authors describe, as an example, the pro- and anti-malignant role of Sema3A. Indeed based on recent findings, Sema3A is instead more a suppressor of tumor progression and an anti-angiogenic molecule (Lavi N, et al Carcinogenesis 2018; Palodetto B et al Stem Cell Res 2016; Jiang H et al, Int J Mol Med 2015; Mishra R et al, Oncogene 2015; Nehil M et al, Oncogene 2014).

Instead, a better and more appropriate example could be Sema3E that is a potent pro-tumoral molecule (Casazza et al J Clin Invest 2010), but it has been described that a cleaved form can be anti-tumor and anti-angiogenic (Casazza et al Embo Mol Med 2012).

In addition, the ref. 46 is not appropriate for the statement.

This sentence should be changes or modified and appropriate ref. should be added.

2. The data described from the authors regarding the role of Sema4A in OVA model of asthma in Sema4A-/- mice and in VEGF tg mice are very interesting, but the final claim in the rows “230-232” is not fully supported by the author’s data and it is more a “speculation” of what Sema4A could do in vivo in cancer progression and in metastatic spreading. Indeed, as the author well described, Sema4A could have anti- or pro-angiogenic/tumoral effects based on the context and on the specific set of Plexins or Nrps used to signal in a specific tumor type. Therefore the author’s statement, based only on the data they presented is not fully appropriate: they should perform further experiments in different mouse models of cancer and dissecting the roles of the different Plexins, before to better verify their conclusions.

The authors should modified (and “stress”) this part adding to their important speculations and interpretations also other statements in which they also “leave open” the issue, highlighting that, depending on the tumor-microenvironment and immune characteristics of different tumor types and on the presence of different Plexins or Nrps, Sema4A could have both pro- or anti tumor/metastatic effect.

Minor comments

a. In the sentences between rows “96-102” the authors should add more appropriate references to support the different and interesting issues discussed of this part.

Author Response

Reviewer #1

Apoorva Iyer & Svetlana P. Chapoval describe very well in this review the role of Sema4A in the regulation of immune system and pathological angiogenesis, presenting both their own data and discussing findings deriving from the literature. This mixed review/article is well written and the authors made a comprehensive overview of the role of Sema4A in several pathological processes. The data the authors presented, the complex relationship between Sema4A and the different Plexins/Nrps receptors and in particular on the role of Sema4A in asthma, inflammation and allergic diseases, are interesting.

However there are some issues and concerns regarding the author’s statements and descriptions of data present in the literature, as well as regarding some of the interpretations of their results, that should be better addressed and modified before publication.

Main concerns:

1. In general along the text and in particular in the conclusions, the role (and “cross-role”) of other Semas in tumor angiogenesis and cancer is not properly described (or clearly), cited and interpreted. They have, at least, to be briefly and better describe it to better put Sema4A in the context of the “complex” system of cancer.

For instance, it has been widely described that different Sema3s (not all well cited in the text) are regulated in tumor angiogenesis, immunity and tumor progression: i) such as the pro-tumoral Sema3E (Luchino J et al Cancer Cell 2013; Casazza et al J Clin Invest 2010; Neufeld G et al, Cell Adh Migr. 2016); ii) Sema3G, as pro-angiogenic (Wallerius M et al, Arterioscler Thromb Vasc Biol 2011); iii) Sema3A, described as endogenous anti-angiogenic and anti-tumor molecule in different mouse model of cancers and human tumors (Luo D et al J Immunol Res. 2018, Gioelli et al Scie. Transl Med 2018, Maione et al., J Clin Invest 2009) and able to revert tumor-associated macrophages (Wallerius M, et al 2016, Cancer Res).

In particular, in the conclusion, when the authors describe some Sema3 as comparison to Sema4A, they should talk in account a more global scenario, to better described the opposite effects observed of Sema4A (and other Semas) in relationship to VEGF and other angiogenic factors.

The authors, in this part of the conclusion should add more appropriate references.

In addition, based on the recent literature, is not fully correct the statement of the rows “259-260” in which the authors describe, as an example, the pro- and anti-malignant role of Sema3A. Indeed based on recent findings, Sema3A is instead more a suppressor of tumor progression and an anti-angiogenic molecule (Lavi N, et al Carcinogenesis 2018; Palodetto B et al Stem Cell Res 2016; Jiang H et al, Int J Mol Med 2015; Mishra R et al, Oncogene 2015; Nehil M et al, Oncogene 2014).

Instead, a better and more appropriate example could be Sema3E that is a potent pro-tumoral molecule (Casazza et al J Clin Invest 2010), but it has been described that a cleaved form can be anti-tumor and anti-angiogenic (Casazza et al Embo Mol Med 2012).

We are very thankful to the Reviewer #1 for all important comments and suggestions. Our answers highlighted in red color are incorporated into the revised manuscript. Indeed, the effect of different semaphorins, especially the class III members, and the relationship between different semaphorin molecules sharing the same receptor in regulating tumor progression present a complex system of cancer development and progression. We believe our clarifications of those roles improved our manuscript significantly.

In addition, the ref. 46 is not appropriate for the statement.

This sentence should be changes or modified and appropriate ref. should be added.

The reference 46 was eliminated from that statement and the statement itself was modified accordingly.

2. The data described from the authors regarding the role of Sema4A in OVA model of asthma in Sema4A-/- mice and in VEGF tg mice are very interesting, but the final claim in the rows “230-232” is not fully supported by the author’s data and it is more a “speculation” of what Sema4A could do in vivo in cancer progression and in metastatic spreading. Indeed, as the author well described, Sema4A could have anti- or pro-angiogenic/tumoral effects based on the context and on the specific set of Plexins or Nrps used to signal in a specific tumor type. Therefore the author’s statement, based only on the data they presented is not fully appropriate: they should perform further experiments in different mouse models of cancer and dissecting the roles of the different Plexins, before to better verify their conclusions.

The authors should modified (and “stress”) this part adding to their important speculations and interpretations also other statements in which they also “leave open” the issue, highlighting that, depending on the tumor-microenvironment and immune characteristics of different tumor types and on the presence of different Plexins or Nrps, Sema4A could have both pro- or anti tumor/metastatic effect.

Indeed, the presented here data are not sufficient to make a conclusion about either pro- or anti-angiogenic/tumoral effects of Sema4A. Considering this point, we modified the manuscript’s title, conclusion, and added more information about the roles of different Plexins and Neuropilins in tumorigenesis and metastasis.

Minor comments

a. In the sentences between rows “96-102” the authors should add more appropriate references to support the different and interesting issues discussed of this part.

The authors discussed and referenced two recently published studies on Sema4A in cancer.

Reviewer 2 Report

The Review titled " Neuroimmune semaphoring 4A in cancer angiogenesis and inflammation" by Iyer and Chapoval is informative and apt for the journal.  It appears that given the controversial role semaphoring 4 as a pro-angiogenic and anti-angiogenic molecule, authors would like to make a case for its anti-angiogenic functions. I have the following comments that need to be addressed.

1.     Title is not informative as to whether it has a positive or a negative role. Authors may clarify the title to increase the readership of the article.

2.     Manuscript overall needs an extensive editing for typographical and grammatical errors in English.

3.     It is not clear whether the figures in the review are a result of experiments carried out in this study or from an earlier study. If it is from this study, then authors must give a more detailed explanation of the experiments especially in the figure legends to make it clear. If they are adapted from an earlier study, authors should cite and acknowledge it.

4.     The expression of Plexin D1 in figure 1 is not convincing and not very significant.

5.     There appears to be a mislabeling of figures and incomplete information in the legends. For example, it is not clear what are I, II and III in figure 3 (labeled figure 2).

6.     Scale bar should be provided for all the microscopy images.

7.     Four-step staining procedure should be cited or method expanded in detail the method.

8.     Authors' use of statistical significance is not standard. For example, p < 0.05 vs p < 0.03 is not very different. Either authors may provide the exact p value determined for the test or state less than p < 0.05 or p < 0.01.

9.     Page 5, Line 209. The Sema4A-VEGFA-FABP4 axis is not clear. Authors may provide a cartoon/figure  or explain in detail.

10. Authors may provide a summary mechanistic cartoon including all the major molecules/players described in the review for better summarizing the role Sema4A in cancer angiogenesis and inflammation.

Author Response

Reviewer #2

The Review titled " Neuroimmune semaphoring 4A in cancer angiogenesis and inflammation" by Iyer and Chapoval is informative and apt for the journal.  It appears that given the controversial role semaphoring 4 as a pro-angiogenic and anti-angiogenic molecule, authors would like to make a case for its anti-angiogenic functions. I have the following comments that need to be addressed.

Title is not informative as to whether it has a positive or a negative role. Authors may clarify the title to increase the readership of the article.

A title has been modified according to reviewer’s comment as indeed it is still unknown based on either our presented data or previously published results if Sema4A has a positive or a negative role in tumorigenesis.

 2.     Manuscript overall needs an extensive editing for typographical and grammatical errors in English.

 Manuscript has been edited to the best abilities of both authors.

It is not clear whether the figures in the review are a result of experiments carried out in this study or from an earlier study. If it is from this study, then authors must give a more detailed explanation of the experiments especially in the figure legends to make it clear. If they are adapted from an earlier study, authors should cite and acknowledge it.

The figures in the review are a result of experiments curried out under NIH R21 R21AI076736 grant. They have not been published before. The authors added a more detailed explanation of the experiments in the figure legends.

 4.     The expression of Plexin D1 in figure 1 is not convincing and not very significant.

 The expression of Plexin D1 is convincing as compared to the isotype control staining. Moreover, Plexin D1 is normally expressed in human vascular tissues during embryonic development. It is absent in adult vasculature unless the certain disease such as, for example, cancer, develops (reference 10 and Gay CM et al., Diverse functions for the semaphorin receptor PlexinD1 in development and disease. Development Biol, 2011, 349:1-19).

5.     There appears to be a mislabeling of figures and incomplete information in the legends. For example, it is not clear what are I, II and III in figure 3 (labeled figure 2).

Roman numerals I, II and III refer to the corresponding stages of cancer progression. It is stated in the figure legend and described in more details in manuscript’s text.

6.     Scale bar should be provided for all the microscopy images.

We show the microscope objective magnification in the figure legends. The pictures were taken in 2010, a quantification of scale bars for those images is not currently possible. Additional information about an image capturing software and a microscope is provided in manuscript’s text.

7.     Four-step staining procedure should be cited or method expanded in detail the method.

 The method is detailed in figure legends.

8.     Authors' use of statistical significance is not standard. For example, p < 0.05 vs p < 0.03 is not very different. Either authors may provide the exact p value determined for the test or state less than p < 0.05 or p < 0.01.

We thank the reviewer for noticing this mistake as not actual SEMs were shown in previous figure. We replaced the panel A of Figure 3 with a new one. For statistical data evaluation we used Student t-test (Microsoft Excel) and Mann–Whitney U test (Prizm-4).

9.     Page 5, Line 209. The Sema4A-VEGFA-FABP4 axis is not clear. Authors may provide a cartoon/figure or explain in detail.

Sema4A is produced in lung DC in response to VEGF overexpression in transgenic mice. Fatty acid binding protein 4 (FABP4) is produced in endothelial cells in response to VEGF exposure including its overexpression in transgenic mice. FABP4 further promotes VEGF-induced lung tissue phenotype including angiogenesis and inflammation (reference 50). Sema4A does not. Additional information about FABP4 pro-angiogenic and pro-inflammatory effects added to the revised manuscript.

10. Authors may provide a summary mechanistic cartoon including all the major molecules/players described in the review for better summarizing the role Sema4A in cancer angiogenesis and inflammation.

A summary mechanistic cartoon is provided (Fig. 1).

Round 2

Reviewer 1 Report

The authors Apoorva Iyer & Svetlana P. Chapoval made different and profound changes and corrections that, overall, improved the manuscript. The news parts and references they added and discussed made this review of high impact for the field and suitable for publication.

Reviewer 2 Report

-